# Towards a Deep Understanding of Multilingual End-to-End Speech Translation

**Haoran Sun, Xiaohu Zhao, Yikun Lei, Shaolin Zhu** and **Deyi Xiong** [*]
College of Intelligence and Computing, Tianjin University, Tianjin, China
{hrsun, zhaoxiaohu, yikunlei, zhushaolin, dyxiong}@tju.edu.cn

## Abstract

In this paper, we employ Singular Value Canonical Correlation Analysis (SVCCA) to analyze representations learnt in a multilingual end-to-end speech translation model trained over 22 languages. SVCCA enables us to estimate representational similarity across languages and layers, enhancing our understanding of the functionality of multilingual speech translation and its potential connection to multilingual neural machine translation. The multilingual speech translation model is trained on the CoVoST 2 dataset in all possible directions, and we utilize LASER to extract parallel bitext data for SVCCA analysis. We derive three major findings from our analysis: (I) Linguistic similarity loses its efficacy in multilingual speech translation when the training data for a specific language is limited. (II) Enhanced encoder representations and well-aligned audio-text data significantly improve translation quality, surpassing the bilingual counterparts when the training data is not compromised. (III) The encoder representations of multilingual speech translation demonstrate superior performance in predicting phonetic features in linguistic typology prediction. With these findings, we propose that releasing the constraint of limited data for low-resource languages and subsequently combining them with linguistically related high-resource languages could offer a more effective approach for multilingual end-to-end speech translation.

## 1 Introduction

Recent years have witnessed the rapid development of end-to-end (E2E) speech-to-text translation (ST) (Berard et al., 2016; Weiss et al., 2017), which has demonstrated remarkable performance and outperformed conventional cascaded systems (Ye et al., 2021; Xu et al., 2021; Han et al., 2021; Ye et al., 2022). The primary advantage of end-to-end ST over cascaded ST is that the new architecture avoids

error propagation and high latency during inference (Sperber and Paulik, 2020).

Recent years have also witnessed that multilingual neural machine translation (NMT) has attracted growing attention (Aharoni et al., 2019; Arivazhagan et al., 2019; Fan et al., 2021; Costa-jussà et al., 2022). One crucial characteristic of multilingual NMT is its knowledge transfer capability, where knowledge learnt from high-resource languages is leveraged to improve translation quality of low-resource languages. Inspired by the success of multilingual NMT, methods used in multilingual NMT have been adapted to multilingual end-to-end speech translation. These include the combination of pre-trained models and fine-tuning (Li et al., 2021) and the incorporation of adapter modules into the encoder/decoder layers (Le et al., 2021). However, due to the limited availability of multilingual ST training data, multilingual E2E ST has been relatively understudied compared to bilingual E2E ST. This not only makes insights and findings into multilingual E2E ST rare, but also leaves many related questions (e.g., in which way methods in multilingual NMT can be successfully adapted to multilingual E2E ST) unanswered. In this paper, our key interest is an in-depth analysis into multilingual E2E ST. We hope findings from the analysis could shed light on its future development.

Previous studies on the interpretability of multilingual pre-trained models (Choenni and Shutova, 2022; Chang et al., 2022a) or multilingual NMT models (Kudugunta et al., 2019) have tried to make the black-box of multilingual models more interpretable by understanding the distribution of language representations learnt by these models. These works have provided valuable insights into multilinguality and have advanced the development of improved multilingual models. Singular Value Canonical Correlation Analysis (SVCCA; Raghu et al., 2017) is a commonly used approach for investigating the representation similarity. It enables us

---

[*]Corresponding author.

to compare the representation similarity obtained through the same data points across different models, layers and languages. SVCCA has been successfully applied to understand the representation of language models and multilingual NMT models.

In this paper, we conduct a comprehensive analysis into a multilingual end-to-end speech translation model trained on 22 languages, utilizing SVCCA as a tool. The research questions we seek to answer are as follows:

- Does multilingual E2E ST demonstrate properties similar to those of multilingual NMT? Specifically, does it exhibit knowledge transfer across languages, benefiting low-resource languages while potentially affecting the performance of high-resource languages?

- What is the distribution of learnt representations like? Do languages from the same language family tend to cluster together based on their learnt sentence representations?

- Can multilingual E2E ST make linguistic typology predictions? Does it demonstrate superior performance in predicting phonetics-related features?

Answering these questions provides insights into multilingual E2E ST. Our findings are as follows:

- The effectiveness of linguistic similarity diminishes when there is insufficient training data for a specific language, which may be attributed to the inadequacy of the training data to support a language-specific sub-space.

- Enhanced encoder representations and aligned audio-text data significantly enhance translation quality, surpassing the performance of bilingual models when the training data is not compromised.

- The encoder representations of multilingual E2E ST exhibit superior performance in predicting phonetic features in linguistic typology prediction.

Based on these observations, we conclude that, for low-resource languages, increasing the amount of parallel training data is more crucial than relying solely on the knowledge transfer ability of the multilingual end-to-end speech translation model. Additionally, building a high-quality language-specific sub-space is crucial for low-resource translation quality.

## 2 Related Work

**End-to-End ST** End-to-end speech-to-text has draw much attention recently due to its lower latency and reduced error propagation compared to traditional cascaded systems (Berard et al., 2016; Weiss et al., 2017). Recent approaches in this field have demonstrated remarkable performance on speech-to-text translation (Vila et al., 2018; Gangi et al., 2019; Zhang et al., 2020a; Wang et al., 2020b,a; Zheng et al., 2021; Chen et al., 2020; Dong et al., 2021; Zhang et al., 2022; Du et al., 2022; Weller et al., 2022; Alastruey et al., 2022; Lam et al., 2022; Lei et al., 2023). However, extending E2E ST to multilingual ST still remains under explored. The first attempt is to develop a multilingual end-to-end ST model based on an LSTM encoder-decoder architecture (Inaguma et al., 2019). Li et al. (2021) propose a combination of Wav2Vec 2.0 (Baevski et al., 2020) and mBART (Liu et al., 2020a), fine-tuning only the layer normalization and multi-head attention layers. Le et al. (2021) insert an adapter layer on the top of each transformer encoder and decoder layer, where only the parameters of the inserted adapters are updated during the fine-tuning stage. Both Di Gangi et al. (2019) and Wang et al. (2021) introduce multilingual speech translation models as baselines along with their proposed speech translation benchmark datasets. Efforts that focus on pre-training for multilingual speech translation, such as XLS-R (Babu et al., 2021), Maestro (Chen et al., 2022), Mu$^2$SLAM (Cheng et al., 2022), Whisper (Radford et al., 2022) and Google USM (Zhang et al., 2023), have been recently explored.

**Multilinguality and Interpretability** The knowledge transfer capability is a crucial aspect of multilingual NMT. It can boost the translation performance of low-resource languages on the one hand while potentially impacting the translation quality of high-resource languages on the other hand (Aharoni et al., 2019; Arivazhagan et al., 2019). Previous approaches on multilingual NMT have focused on designing efficient language-specific modules (Bapna and Firat, 2019; Philip et al., 2020; Zhang et al., 2020b; Zhu et al., 2021; Zhang et al., 2021; Lin et al., 2021) or leveraging linguistic similarity among languages (Sachan and Neubig, 2018; Tan et al., 2019; Oncevay et al., 2020; Sun and Xiong, 2022; Baziotis et al., 2022) to strike a balance in this trade-off. Understanding

the inner workings of multilingual models remains an intriguing question (Conneau et al., 2020; Rama et al., 2020; Liang et al., 2021). Singular Value Canonical Correlation Analysis (SVCCA; Raghu et al., 2017) has been used to quantify the similarity between sets of representations in various language-related models, including language models (Saphra and Lopez, 2019), multilingual language models (Chang et al., 2022b), multilingual NMT models (Kudugunta et al., 2019; Oncevay et al., 2020) and end-to-end ASR models (Ollerenshaw et al., 2022). Our paper is most similar to the study conducted by Kudugunta et al. (2019), as they utilize SVCCA to analyze the distribution of languages in multilingual NMT with sentence-level representations. They achieve insights into language similarity, which facilitate succeeding studies on multilingual NMT. Wang et al. (2023) use t-SNE (Van der Maaten and Hinton, 2008) to analyze the representations learnt by Maestro (Chen et al., 2022), but their interest is in the impact of alignment between speech and text modality on speech translation, while we focus on the multilinguality analysis in multilingual E2E ST.

## 3 Empirical Anlysis Setup

### 3.1 Data and Model

We study multilingual end-to-end speech translation using the CoVoST 2 dataset [1] (Wang et al., 2021), which is an English-centric dataset that supports translation from English to 15 languages (En→X) and translation from 21 languages to English (X→En). Among the X→En directions, only 4 languages have more than 100 hours of training data, while the remaining 17 languages have limited training resource, with less than 50 hours available. We categorize the languages in X→En directions into high/mid/low-resource languages according to the amount of training data available for them. Specifically, the high-resource languages include French, German, Spanish and Catalan, the mid-resource languages contain Persian, Italian, Russian, Portuguese, Chinese and Dutch, the remaining languages are considered as low-resource languages.

In order to accurately calculate similarity between languages based on sentence-level representations, it is crucial to minimize the impact of different meanings of sentences, which may introduce

confounding factors (Kudugunta et al., 2019). In our study, we mitigate this issue by selecting a set of semantically similar sentences for each pair of languages. To accomplish that, we utilize LASER[2] (Schwenk and Douze, 2017; Heffernan et al., 2022) to mine parallel sentences for each pair of given languages as evaluation datasets to measure similarity across different languages. More details are provided in Appendix A.

Analysis experiments are conducted on three directions: En→X, X→En and X→X. We tokenize the translated texts using a jointly learnt unigram Sentencepiece model[3] (Kudo and Richardson, 2018) with a vocabulary size of 10K for each directions. As for the audio data, we extract 80-dimensional log mel-scale filter bank features (windows with 25ms size and 10ms shift).

To train the multilingual E2E ST models, we first train separate multilingual ASR models for each translation directions. We then use the trained multilingual ASR encoder to initialize the encoder of the multilingual ST models. Following Arivazhagan et al. (2019), we use the temperature-based sampling method during training X→En and X→X models with a temperature value of $T = 5$ to alleviate the heavy imbalance between language pairs.

As for the bilingual ST models, we adopt the settings used by Wang et al. (2021).

For evaluating, we report case-sensitive detokenized BLEU using SacreBLEU (Post, 2018) except for English-Chinese and English-Japanese where we use the tokenizer provided by SacreBLEU (zh for Chinese and ja-mecab for Japanese). All models are implemented with the Fairseq toolkits[4] (Ott et al., 2019).

More details are provided in Appendix B.

### 3.2 SVCCA

We employ Singular Value Canonical Correlation Analysis (SVCCA; Raghu et al., 2017) for our analysis. SVCCA is a method that allows us to compare the correlation between two vector representations. It is invariant to affine transformations and fast to compute. To apply SVCCA, we consider a set of data points containing $N$ examples. The representation of a layer can be regarded as the hidden states of the layer of these $N$ data points. Let $l_1 \in \mathbb{R}^{N \times D_1}$ and $l_2 \in \mathbb{R}^{N \times D_2}$ denote the representations of two layers, where $D_1$ and $D_2$ are the

---

[1] https://github.com/facebookresearch/covost

[2] https://github.com/facebookresearch/LASER
[3] https://github.com/google/sentencepiece
[4] https://github.com/facebookresearch/fairseq

| LANGs | Hours | Bi ST (X→En) | X→En | X→X (X→En) | Bi ST (En→X) | En→X | X→X (En→X) |
|-------|-------|--------------|------|------------|--------------|------|------------|
| fr | 264 | 26.47 | 26.67 | 27.99 | - | - | - |
| de | 184 | 17.69 | 18.06 | 20.14 | 16.03 | 19.42 | 17.99 |
| ca | 136 | 19.28 | 23.04 | 24.33 | 21.64 | 25.06 | 23.71 |
| es | 113 | 23.13 | 27.46 | 29.10 | - | - | - |
| fa | 49 | 3.83 | 3.27 | 3.17 | 12.78 | 16.50 | 15.69 |
| it | 44 | 11.19 | 20.30 | 20.87 | - | - | - |
| ru | 18 | 14.77 | 17.39 | 15.97 | - | - | - |
| pt | 10 | 6.13 | 10.16 | 8.23 | - | - | - |
| zh | 10 | 5.68 | 6.37 | 7.41 | 23.67 | 29.63 | 28.53 |
| nl | 7 | 3.04 | 3.65 | 5.32 | - | - | - |
| tr | 4 | 3.51 | 3.57 | 3.42 | 10.09 | 12.50 | 11.31 |
| et | 3 | 0.47 | 0.95 | 0.89 | 12.93 | 15.81 | 14.31 |
| mn | 3 | 0.22 | 0.36 | 0.20 | 9.43 | 11.78 | 10.84 |
| ar | 2 | 4.31 | 2.29 | 1.08 | 12.22 | 14.14 | 12.94 |
| cy | 2 | 2.56 | 2.82 | 2.74 | 23.88 | 26.32 | 25.19 |
| lv | 2 | 2.51 | 1.93 | 1.03 | 13.10 | 15.42 | 14.03 |
| sl | 2 | 2.97 | 3.39 | 1.76 | 15.97 | 18.92 | 16.97 |
| sv | 2 | 3.24 | 1.38 | 1.14 | 21.77 | 25.07 | 23.58 |
| ta | 2 | 0.31 | 0.09 | 0.12 | 10.92 | 13.63 | 12.73 |
| id | 1 | 2.39 | 0.87 | 0.24 | 20.24 | 23.55 | 23.14 |
| ja | 1 | 1.70 | 1.36 | 0.24 | 20.73 | 25.23 | 24.17 |
| avg | - | 7.40 | 10.33 | 10.23 | 16.36 | 19.53 | 18.34 |

Table 1: Analysis results on the CoVoST 2 dataset. We compare results of multilingual end-to-end speech translation trained on three different translation directions. Hours denote the total number of hours of training audio data for the language on the source side. Bi ST is the bilingual end-to-end speech translation model.

dimensions of the layers corresponding to $l_1$ and $l_2$, respectively. SVCCA proceeds as follows:

1. Perform Singular Value Decomposition (SVD) on $l_1$ and $l_2$ to get sub-spaces $l_1' \subset l_1$, $l_2' \subset l_2$ which comprise of the most important directions of the original $l_1$ and $l_2$, where $l_1' \in \mathbb{R}^{N \times D_1'}, l_2' \in \mathbb{R}^{N \times D_2'}$. We retain enough dimensions to keep 99% of the variance in the data.

2. Use Canonical Correlation Analysis (CCA) to project $l_1'$ and $l_2'$ onto a shared subspace, i.e., computing $\tilde{l}_1 = W_X l_1'$, $\tilde{l}_2 = W_Y l_2'$ to maximize the correlations corrs $= \{\rho_1, \cdots, \rho_{\min(D_1', D_2')}\}$ between the new subspaces.

We follow Raghu et al. (2017) to use the mean of the correlations:

$$\bar{\rho} = \frac{1}{\min(D_1', D_2')} \sum_i \rho_i \qquad (1)$$

Following Kudugunta et al. (2019), we adopt the sequence-based SVCCA which involves performing SVCCA on the output of the layer and averaging the results over sequence time-steps. This sequence-based SVCCA can compare the unaligned sequences across different languages in a more suitable way than the original token-level strategy.

## 4 Main Results

We present the analysis results on the CoVoST 2 dataset in Table 1. Surprisingly, the multilingual E2E ST model exhibits a distinct pattern compared to the multilingual NMT model. In the X→En translation direction, which includes high/mid/low-resource source languages, the multilingual ST model does not demonstrate the same knowledge transfer ability observed in the multilingual NMT model. In general, the low-resource languages do not benefit from the high-resource languages and continue to exhibit low translation quality, even when the low-resource language is linguistically related to a high-resource language. For instance, Swedish (sv), which is from the Germanic language branch of Indo-European language family, shares linguistic similarities to German (de) and Dutch (nl), both of which are Germanic languages and considered high/mid-resource languages. However, Swedish does not benefit from German and still exhibits poor translation quality.

However, on the other hand, all of the high-resource languages gain significant improvements in translation quality, even surpassing the performance of the corresponding bilingual ST models. This phenomenon is particularly evident in the En→X translation direction, where all the languages can be considered high-resource. The mul-

tilingual system outperforms the bilingual systems by an average BLEU of 3.17. Similarly, in the X→En translation direction, the high-resource languages (French, German, Catalan, and Spanish) also demonstrate improved performance with an average BLEU of 23.81, compared to the average BLEU of 21.64 achieved by the bilingual systems.

From these comparisons, we can draw the following conclusions: (I) Improved audio representations enhance the information encoding capability of the encoder component, resulting in better translation quality for high/mid-resource languages. In the X→En direction, the multilingual training audio data enhances the encoder's ability by providing a more suitable encoding space, thereby boosting the performance for high-resource languages. (II) The introduction of well-aligned audio-text data also benefits speech translation quality. In the En→X direction, although the total amount of audio data remains the same as in the bilingual setting, aligning this data with multilingual text helps the model learn better alignment between audio and text. This phenomenon has been extensively studied in the context of end-to-end bilingual speech translation, where it is referred to as the modality gap (Liu et al., 2020b; Han et al., 2021; Ye et al., 2022; Fang et al., 2022). (III) The performance of low-resource languages is still limited by the availability of aligned audio-text data. This scarcity of data hampers the model's ability to capture the nuances and specific characteristics of low-resource languages, leading to lower translation quality compared to high/mid-resource languages.

For the X→X model trained on all translation directions, it still surpasses bilingual systems for all high-resource languages. However, it falls behind the En→X model when translating from English to other languages. We conjecture that this performance gap is due to the limited model capacity, where related parameters are affected by interference from mid- and low-resource languages in the X→En direction. Interestingly, the performance on the high-resource languages in the X→En direction outperforms the model trained only in the X→En direction. This behavior is reminiscent of the behavior observed in multilingual NMT models, where the performance of high-resource languages is normally sacrificed to benefit low-resource languages. In this case, the high-resource languages correspond to the languages in the En→X direction, while the low-resource languages are the languages

in the X→En direction, which have sufficient training data. However, the languages in the X→En direction with only few hours of audio data still achieve low translation quality.

These results provide support for the observation that the multilingual ST model exhibits a different pattern compared to the multilingual NMT model. The multilingual ST model leverages better audio representations and alignments between audios and texts to achieve improved translation quality for languages that have sufficient parallel training data. However, the linguistic aspect loses its effectiveness in the multilingual ST model, as low-resource languages are constrained by the limited amount of training data and do not benefit from their linguistically related high-resource languages. This finding aligns with what we observe in Section 5.1. Overall, the amount of parallel training data is more crucial than the linguistic relatedness of languages for the performance of the multilingual ST model.

## 5 Language Similarity

To thoroughly examine the impact of language similarity on the multilingual E2E ST model trained in the X→X direction, we conducted SVCCA analysis on our LASER-mined evaluation datasets with semantically similar sentences, as mentioned in Section 3.1. It's worth noting that the LASER-mined evaluation datasets are created specifically for the X→En direction, as the CoVoST 2 dataset already provides a multi-way-parallel test set for the En→X direction, where the English audio remains the same across all languages. The SVCCA scores are computed based on layer-wise hidden states of the encoder in the X→En direction and the decoder in the En→X direction. This analysis allows us to examine the similarity between languages across different layers of the model.

In the upcoming sections, our discussion will center around language similarity from multiple perspectives. We consider language family as an important factor for analysis, e.g., Indo-European, which encompasses a significant portion of the languages in our dataset. We also examine language branch, like Romance and Germanic. Additionally, the amount of available training data for each language is taken into account during our analysis. Finally, we delve into patterns related to the writing systems of languages on the decoder side for translation generation. By exploring these aspects,

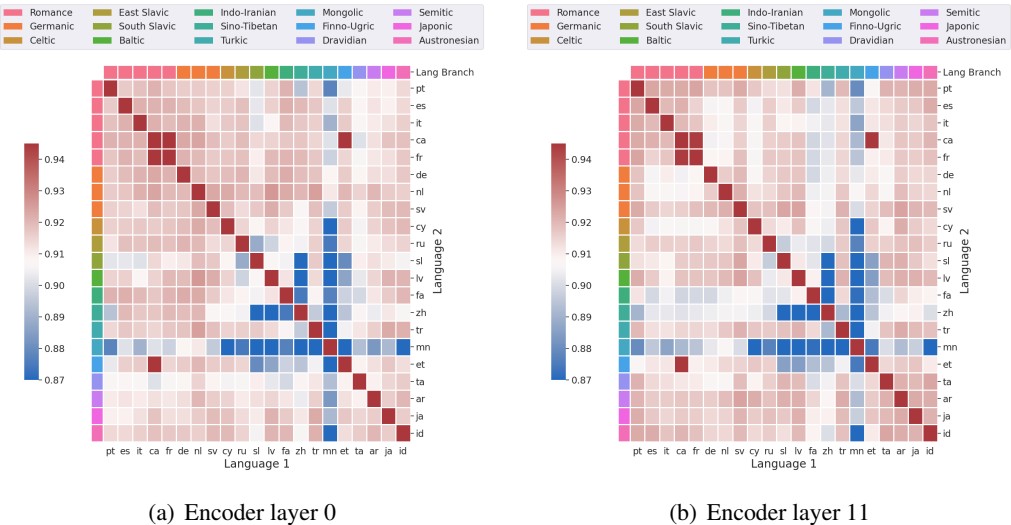

| (a) Encoder layer 0 | (b) Encoder layer 11 |

Figure 1: SVCCA scores between the representations (encoder layer 0 and encoder layer 11) of X→En language pairs (i.e., pairs of X), which is calculated on our LASER-mined evaluation datasets. Red cells indicate that the two languages are more related to each other (higher SVCCA scores) and blue cells indicate that the two languages are less related (lower SVCCA scores). Best viewed in color.

we aim to gain insights into how linguistic factors, training data availability, and written script characteristics influence the performance of multilingual E2E ST.

## 5.1 SVCCA Scores of the Source Languages in X→En Translation

We first visualize SVCCA scores of language pairs in Figure 1. We only demonstrate the results of the first encoder layer (encoder layer 0 in Figure 1(a)) and the last encoder layer (encoder layer 11 in Figure 1(b)) to show a comparison between them, the results of other layers are provided in Appendix D.1.

From the comparison between the results of encoder layer 0 and encoder layer 11, we observe a decrease in the mean SVCCA scores (0.89 Vs. 0.86) for each pair of languages. This decrease suggests that languages tend to utilize their language-specific parameters as layers go deep. In the first layer (layer 0), languages exhibit more general representations, as evidenced by the relatively close SVCCA scores of different language pairs. However, as we go deep to the final layer (layer 11), we observe a clear tendency for SVCCA scores to converge in terms of language branches. Specifically, the Romance languages demonstrate higher SVCCA scores among themselves compared to languages from other language branches, indicating a stronger similarity in their representations.

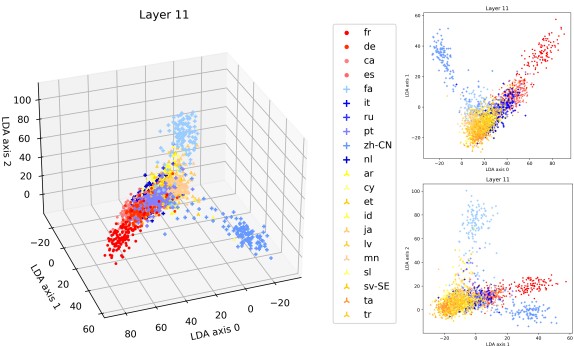

Figure 2: Representations of encoder layer 11 projected onto a linear sub-space with three LDA axes. The projection on the top right corner visualizes the sub-space with LDA axis 0 and 1. The projection on the bottom right corner visualizes the sub-space with LDA axis 1 and 2. Red/blue/yellow-colored items are high/mid/low-resource languages, respectively.

Similarly, the Germanic languages exhibit a similar pattern, with higher SVCCA scores observed within this language branch.

We observe an interesting phenomenon among the mid-resource languages (Persian, Italian, Russian, Portuguese and Dutch) that are all from the Indo-European language family. In Figure 1(b), we can see that Portuguese, Italian, Russian and Dutch exhibit higher SVCCA scores with other languages in the Indo-European language family. These languages also demonstrate better translation quality in the multilingual ST model compared

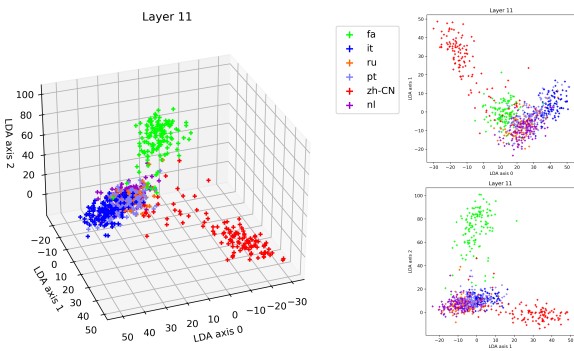

Figure 3: Representations of encoder layer 11 projected onto a linear sub-space with three LDA axes, exclusively comprising mid-resource languages. The projection on the top right corner visualizes the sub-space with LDA axis 0 and 1. The projection on the bottom right corner visualizes the sub-space with LDA axis 1 and 2.

to their bilingual counterparts. However, Persian is not similar to other languages, as indicated by lower SVCCA scores across all languages in the Indo-European language family. This low similarity leads to poor translation quality in the multilingual ST model, despite that Persian has 49 hours of training data available.

In the case of low-resource languages, we have observed that they tend to exhibit higher SVCCA scores with most languages, indicating similarity in their representations. However, despite this similarity, these low-resource languages still suffer low translation quality in the multilingual ST model.

In order to thoroughly examine multilingual ST for low-resource languages, we utilize the linear discriminant analysis (LDA) to explicitly identify language-specific sub-spaces, as described in previous works (Liang et al., 2021; Chang et al., 2022b). The results are visualized in Figure 2. It can be observed from Figure 2 that high- and mid-resource languages exhibit distinct language-specific sub-spaces along different LDA axes, whereas low-resource languages remain within a common sub-space without a prominent language-specific distribution. Due to the lack of sufficient training data, low-resource languages are unable to develop their own language-specific sub-spaces within the encoder. Instead, they mainly locate on a shared sub-space that is common to all languages. The absence of language-specific sub-spaces undermines translation performance for these low-resource languages. This finding aligns with the findings of previous studies on multilinguality with a fairness lens (Wu and Dredze, 2020; Choudhury and

Deshpande, 2021; Cabello Piqueras and Søgaard, 2022).

With the help of LDA, we can illustrate the reasons why Persian differs from other Indo-European languages and does not benefit from linguistic similarities. In Figure 3, we present a simplified version of Figure 2, which contains only six languages. From Figure 3, we can discern that Persian occupies a distinct position along the LDA axis compared to the other Indo-European mid-resource languages. Similarly, Chinese exhibits a similar pattern, primarily due to its affiliation with Sino-Tibetan languages. We believe that this variance in the LDA axis can also be interpreted as a distinct language-specific subspace, which contributes to the challenges in transferring translation abilities to Persian languages.

Based on the analysis conducted on the encoder side, we identify two factors that affect translation quality: language similarity and the amount of training data. Language similarity impacts the level of knowledge transfer across languages which contributes to the development of high-quality representations. And the quantity of training data plays a crucial role in establishing language-specific sub-spaces, which is also vital for translation quality.

## 5.2 SVCCA Scores of the Target Languages in En→X Translation

We visualize the SVCCA scores on the decoder side for 15 languages in Figure 4. We present the results of the first decoder layer (decoder layer 0) and the last decoder layer (decoder layer 5) to compare their SVCCA scores. The results of other layers are displayed in Appendix D.2.

When comparing the SVCCA scores of decoder layer 5 with layer 0, we observe an increase in the mean SVCCA scores. This gradual increase indicates that, during the translation process from English to other languages, different languages tend to utilize a more general sub-space at the top decoder layer.

Given that the modality of the decoder side is text, we also observe a pattern related to the writing system correlation in the SVCCA score results. In Figure 4(b), we observe higher SVCCA scores for Chinese and Japanese compared to other languages. Although Chinese and Japanese have distinct writing systems, Japanese borrow characters from Chinese (e.g. Kanji). We also notice a similar pattern between Arabic and Persian, both of which use the

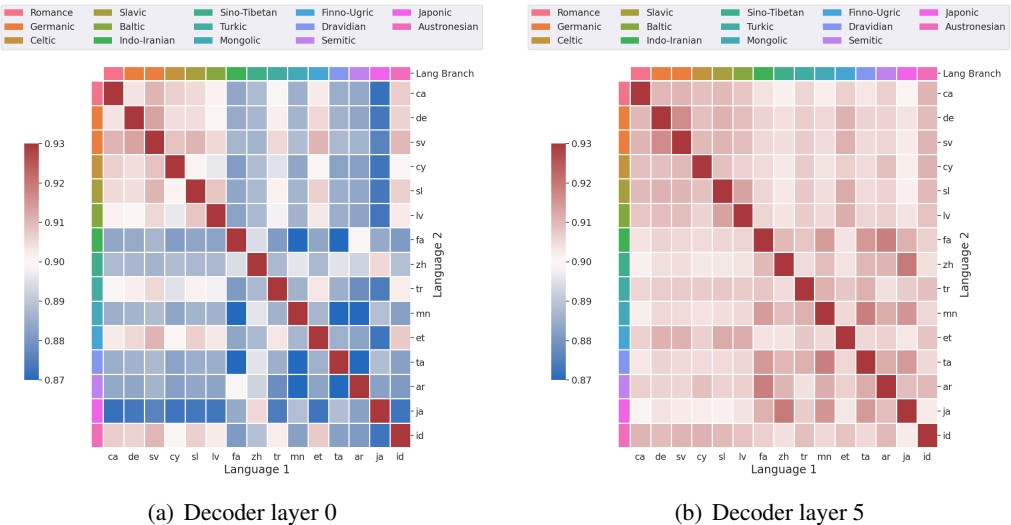

(a) Decoder layer 0        (b) Decoder layer 5

Figure 4: SVCCA scores between the representations (decoder layer 0 and decoder layer 5) of En→X language pairs. Red cells indicate that the two languages are more related to each other (higher SVCCA scores) while blue cells indicate that the two languages are less related (lower SVCCA scores). Best viewed in color.

Arabic alphabet. This may explain that the SVCCA scores for Arabic and Persian are higher compared to those with other languages. This finding also in line with the finding of Kudugunta et al. (2019).

## 6 Linguistic Typology

We conduct experiments on the linguistic typology prediction for our multilingual end-to-end speech translation model trained on the X→X direction on the CoVoST 2 dataset.

**Dataset** We employ typological features from URIEL typological database[5] (Littell et al., 2017) for experiments. URIEL is a typological compendium which accommodates diverse linguistic resources from several typological databases such as WALS (Dryer and Haspelmath, 2013), PHOIBLE (Moran and McCloy, 2019), Ethnology (Lewis et al., 2015) and Glottolog (Hammarström et al., 2021). We used lang2vec library to query URIEL database which provides uniform interface to access various linguistic features. We mainly use syntax, phonology and phonetic inventory typological features in our work.

**Prediction Methods** We adopt two methods commonly used in previous studies: k-nearest neighbors approach ($k$-NN) and logistic regression (Malaviya et al., 2017; Oncevay et al., 2020). We utilize the averaged sentence representations

[5]https://www.cs.cmu.edu/~dmortens/projects/07_project

| Feature | $k$-NN | | | | | Logistic |
| | 1 | 3 | 5 | 7 | Max | |
|---|---|---|---|---|---|---|
| Syntax | 76.13 | 78.74 | 80.60 | **81.05** | 81.05 | 78.62 |
| Phonology | 87.74 | 90.31 | **91.64** | 91.20 | 91.64 | 87.96 |
| Inventory | 83.87 | 87.88 | 88.20 | **88.53** | 88.53 | 85.78 |

Table 2: Linguistic typology test accuracy on syntax, phonology and phonetic inventory features using the language representations learnt by the encoder. $k$ denotes the number of nearest neighbors in $k$-NN. Max denotes the maximum accuracy when $k$ varies in 1, 3, 5, 7.

obtained from the encoder for all samples on the CoVoST 2 test set. These representations serve as vector representations of each language, which we employ for probing the typological features. In the $k$-NN approach, we set $k$ as odd numbers and vary $k$ in $\{1, 3, 5, 7\}$. We leave one language out and take samples of the remaining languages as training data to predict the linguistic typology feature. This step is repeated for all languages, and we report the average prediction accuracy across all languages.

**Results** We present prediction results in Table 2. A notable observation is that the vector representations derived from our multilingual E2E ST model demonstrate higher accuracy in predicting phonology features and phonetic inventory features compared to syntax features. This finding aligns with expectations since the vector representations are obtained through the encoder, which primarily processes the audio modality and is more likely

to capture phonological and phonetic information. It suggests that the multilingual E2E ST model effectively encodes and represents phonological aspects of languages in its sentence representations. However, the lower accuracy in predicting syntax features indicates that it may be difficult for the model to capture syntax-related information solely from the audio modality.

# 7 Discussion

Our study reveals several key observations that might be of interest to researchers and practitioners working in multilingual E2E ST.

Our analysis identifies two factors that affect multilingual speech translation quality: linguistic similarity and the amount of training data. The availability of an adequate amount of training data is crucial for the model to learn language-specific sub-spaces, particularly for low-resource languages. Linguistic similarity can facilitate knowledge transfer across languages, especially when low-resource languages are not constrained by limited data. Thus, addressing data scarcity in the multilingual setting and improving the representation space for individual languages should be explored to enhance multilingual translation quality. In another perspective, learning a high-quality representation space for a low-resource language with limited data is also a path to achieve linguistic fairness of multilingual E2E ST.

It is widely recognized that enhancing audio representations can improve the performance of the acoustic model, and this principle applies to multilingual/bilingual E2E ST models as well.

# 8 Conclusion

In this study, we have analyzed language representational similarity learnt by a multilingual end-to-end speech translation model trained on 22 languages via SVCCA. Through our analysis, we have findings that shed light on the performance of such models. We observe that the amount of available data plays a significant role in limiting the effectiveness of knowledge transfer across languages in multilingual speech translation. Using SVCCA to evaluate the similarity across languages, we observe a clustering effect in terms of language branch, indicating aggregation of linguistic features within language families. We also use learnt language representations to probe linguistic typology and find that the multilingual ST model performs better

on phonetic-related features compared to syntax features.

# Limitations

The main limitations of this study lie in two aspects: the quality of the LASER-based evaluation datasets and the analysis perspective.

In our approach, we set the threshold of 1.05 to determine semantic equivalence, which is lower than the standard threshold of 1.2 typically used to ensure high-quality aligned bitext. Consequently, there is a possibility that some of the sentences identified as parallel text may possess different meanings, potentially introducing a confounding factor that could impact the reliability of our analysis results based on SVCCA, since the scarcity of audio data is much more serious than text data.

Regarding the analysis perspective, our investigation on multilingual end-to-end speech translation focuses on the linguistic aspect. However, we have not conducted an analysis of the model from the phonological perspective, which has the potential to offer additional insights into multilingual E2E ST. Unfortunately, due to the absence of a reliable standard mapping between audio and phoneme, conducting a comprehensive analysis, particularly for low-resource languages, poses significant challenges.

# Ethics Statement

This study presents an analysis of multilingual end-to-end speech translation, primarily based on the CoVoST 2 dataset commonly used in speech translation. Additionally, we incorporate the Common Voice project to construct our evaluation sets, which is released under the Creative Commons Attribution Share-Alike 3.0 Unported license (CC BY-SA 3.0).

# Acknowledgments

The present research was supported by the Key Research and Development Program of Yunnan Province (No. 202203AA080004) and the Natural Science Foundation of Xinjiang Uygur Autonomous Region (No. 2022D01D43). We would like to thank the anonymous reviewers for their insightful comments.

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

## A  LASER-mined Evaluation Datasets

The dataset CoVoST 2, which is based on the Common Voice project (Ardila et al., 2020) with Version 4, has a limited amount of audio training data for most languages in the X→En directions. This limitation poses a challenge on the selection of semantically similar sentences as evaluation datasets for low-resource languages. To overcome this challenge, we incorporate Common Voice version 13 as supplementary data for CoVoST 2 specifically for the low-resource languages. We filter out the audio data from version 13, which is duplicated with the CoVoST 2 training set, reserving the remaining data as a new evaluation set for each corresponding language. Subsequently, we utilize LASER to mine bitext (transcription of audio) between each pair of given languages. A threshold of 1.05 is set to determine the extracted bitext, and this process is repeated for each language pair within all the languages in our study. These extracted evaluation datasets are then used to measure similarity across different languages.

## B  Training Details

We used the Transformer (Vaswani et al., 2017) as the backbone for our multilingual end-to-end speech translation model, which has 12 layers for the encoder and 6 layers for the decoder with 16 attention heads and 1024 dimensions for embeddings, 4096 dimensions for FFNs. We set the dropout rate to 0.3 for the multilingual models. We initialized the transformer encoder with a pre-trained ASR model, which shares the same configuration as our multilingual ST model. We trained different ASR models for different translation directions, e.g., an English ASR model for the English→X direction, a multilingual ASR model (contains 21 languages) for the X→English direction and a multilingual ASR model (contains 21 languages + English) for the X→X direction.

We appended a language token at the beginning of the translated sentences to denote which language should be translated to following Johnson et al. (2017). We did not add any language-specific token or embedding at the source side for multilingual ASR and ST models.

We optimized parameters using Adam optimizer (Kingma and Ba, 2015) with a label smoothing rate of 0.1. The learning rate was scheduled according to the inverse square root of running steps with a warm-up step of 2500. We adopted the early stop-

| LANGs | WER | LANGs | WER |
|-------|-----|-------|-----|
| en | 23.14 | tr | 47.73 |
| fr | 16.70 | et | 58.83 |
| de | 18.94 | mn | 60.78 |
| ca | 11.33 | ar | 58.36 |
| es | 13.13 | cy | 61.66 |
| fa | 72.88 | lv | 48.56 |
| it | 20.47 | sl | 48.28 |
| ru | 30.55 | sv | 60.82 |
| pt | 32.63 | ta | 73.31 |
| zh | 38.08 | id | 47.39 |
| nl | 50.22 | ja | 47.59 |

Table 3: Results of the multilingual ASR model which is used to train the multilingual ST model in the X→X translation directions.

ping strategy with patience set to 5 for English→X and X→X model and averaged the last 5 checkpoints for inference. As for the X→English model, we set the maximum number of updates to 10K and averaged 5 checkpoints for inference, we chose the best averaged model according to the average BLEU on the validation sets and then evaluated it on the test sets.

## C  Results of Multilingual ASR

In Table 3, we present the results of our multilingual ASR model. A consistent pattern emerges from these results, mirroring the findings of the multilingual ST model in the X→En translation directions. In these cases, the low-resource languages continue to suffer low performance, even in the context of ASR tasks.

## D  Language Similarity

### D.1  SVCCA Scores of the Source Languages in X→En Translation

Figure 5 shows SVCCA scores of language pairs in the X→En direction across all encoder layers (from layer 0 to layer 11).

### D.2  SVCCA Scores of the Target Languages in En→X Translation

Figure 6 shows SVCCA scores language pairs in the En→X direction across all decoder layers (from layer 0 to layer 5).

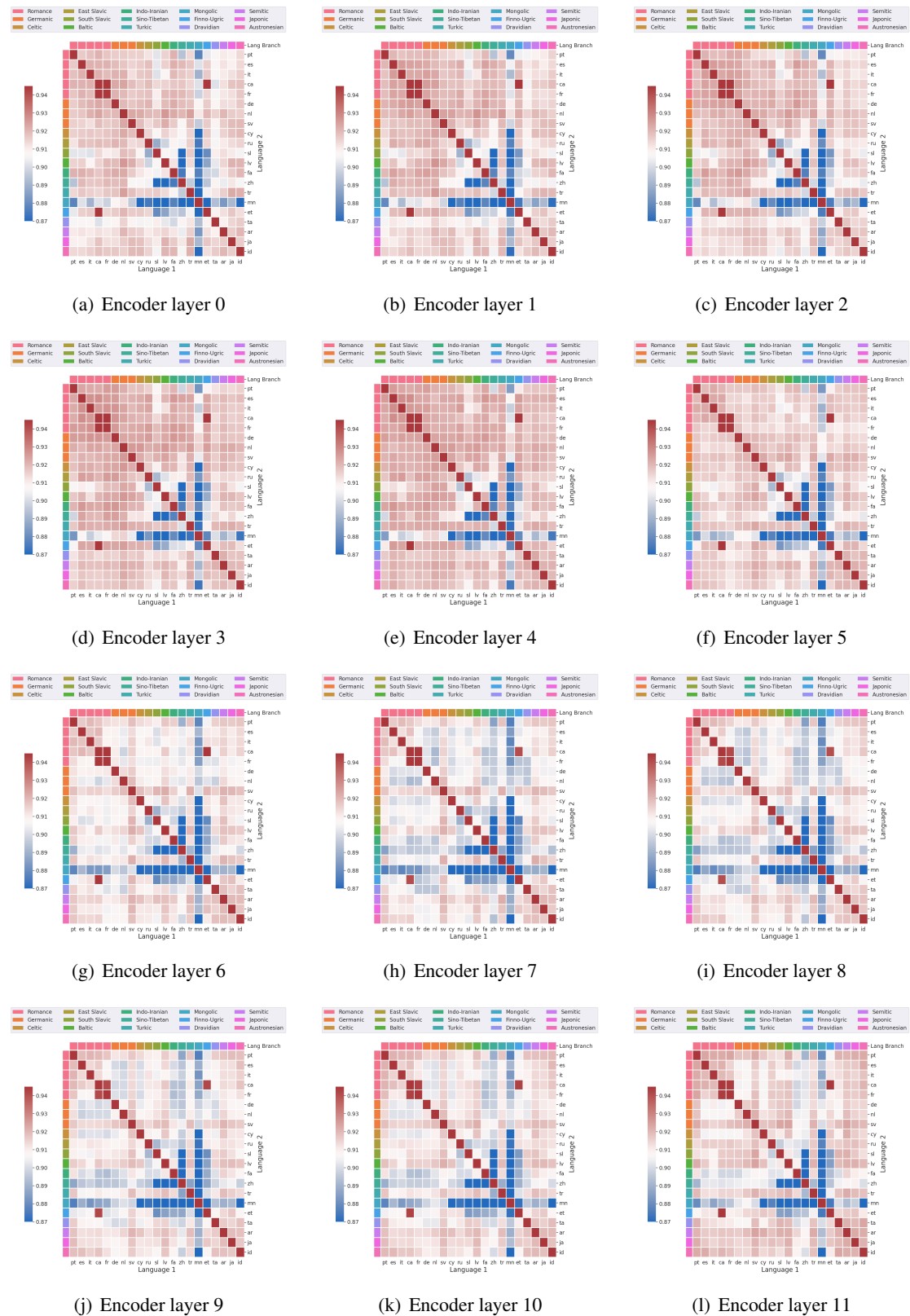

Figure 5: SVCCA scores between the representations of X→En language pairs (i.e., pairs of X) across all encoder layers, which is calculated on our LASER-mined evaluation datasets. Red cells indicate that the two languages are more related to each other (higher SVCCA scores) and blue cells indicate that the two languages are less related (lower SVCCA scores). Best viewed in color.

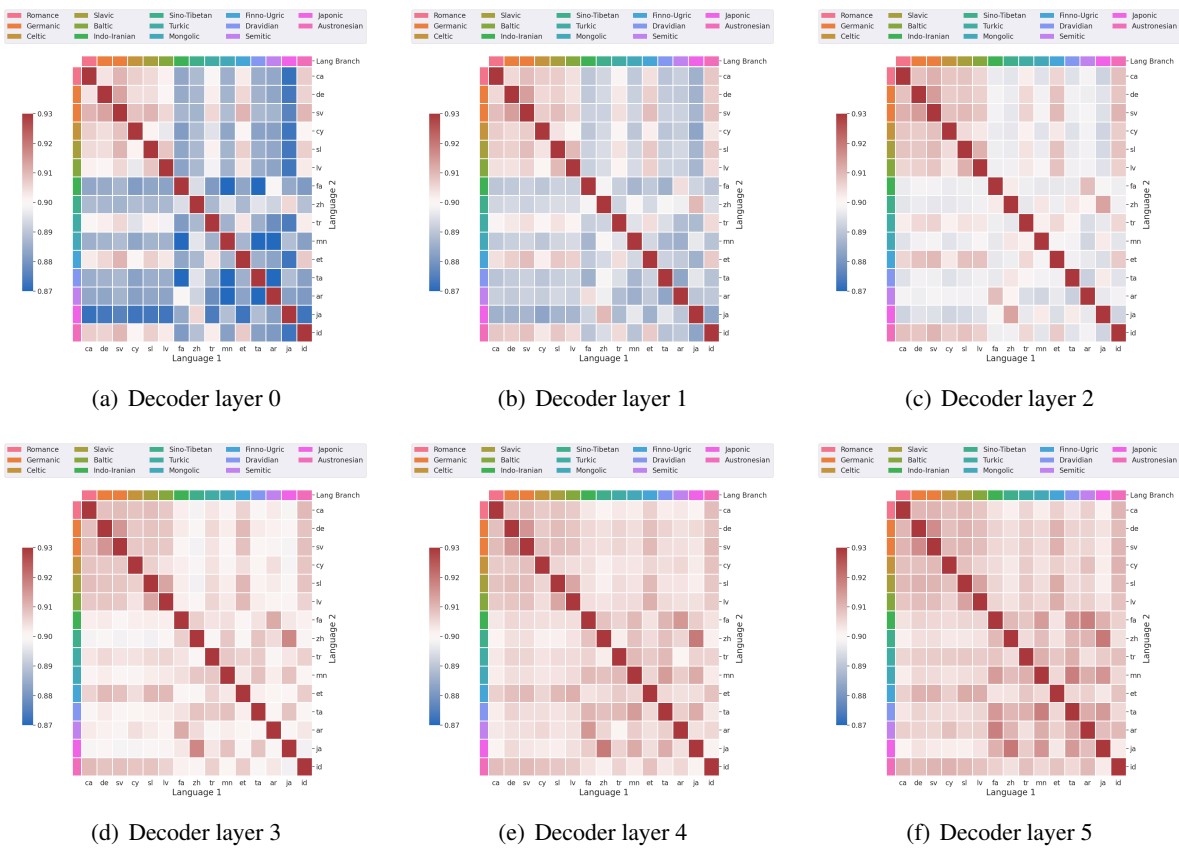

Figure 6: SVCCA scores between the representations of En→X language pairs across all decoder layers. Red cells indicate that the two languages are more related to each other (higher SVCCA scores) while blue cells indicate that the two languages are less related (lower SVCCA scores). Best viewed in color.