# OpenReview forum: "Towards a Deep Understanding of Multilingual End-to-End Speech Translation"
_EMNLP/2023/Conference — EMNLP 2023 Findings_

### Official Review · Reviewer_JjpA · 2023-08-04

**Soundness:** 3

**Excitement:**

4: Strong: This paper deepens the understanding of some phenomenon or lowers the barriers to an existing research direction.

**Paper Topic And Main Contributions:**

This paper analyzed a multilingual end-to-end speech translation (E2E ST) model using Singular Value Canonical Correlation Analysis (SVCCA), with a focus on the learned representations, both from the perspective of language similarity and linguistic typology.
The authors found that linguistic similarity may be less effective when there is insufficient training data in a language. They also found that the encoder representations present better accuracy for phonology/phonetic features.

**Questions For The Authors:**

Did the authors try to perform ablation studies by, for instance, train multilingual E2E ST models with languages grouped by language family/branch, rather than using all the languages available?

**Reasons To Accept:**

The paper is very clearly written and easy to follow.

Analysis of multilingual representations in E2E ST is lacking in literature, and though some of the conclusions are not very surprising (ex. having more training data is better than just using knowledge transfer), I believe this empirical investigation work is a great contribution providing insights for better understanding of the multilingual representations in ST systems, for further multilingual ST research & also for practitioners when allocating resources for low-source languages.

I also appreciate the analysis made from different perspectives (linguistic factors, amount of training data, characteristics of writing systems, etc.).

**Reasons To Reject:**

Given the limitations in terms of diversity, scale and quality for some language pairs (ex. on some low-resource pairs, there is only 1 hour of data), it remains unclear to which extent the conclusions will remain valid. However it is not quite the focus of contribution of this paper, more a limitation of data available, so I don't consider this point as a strong reason for rejection, rather some possible directions for improvements.

Some analysis on language similarity are a bit lightweight and the di/similarity between languages are not very discussed (for instance, how Persian is dissimilar compared to other languages in the dataset from the Indo-European?), making it uneasy to disentangle the factors between language similarity and training data availability. Some finer-grained analysis on the clusters identified in Figure 2 or some ablation studies could be helpful for a deeper understanding of the impacts of the factors the authors investigated.

**Reproducibility:**

4: Could mostly reproduce the results, but there may be some variation because of sample variance or minor variations in their interpretation of the protocol or method.

**Reviewer Confidence:**

4: Quite sure. I tried to check the important points carefully. It's unlikely, though conceivable, that I missed something that should affect my ratings.

**Typos Grammar Style And Presentation Improvements:**

Typos:
- Figure 2, caption: lacks a period mark at the end.

Grammar:
- Last paragraph of 5.2, "This finding also in line with the finding Kudugunta et al. (2019)." -> "[...] with the finding of [...]"

Style:
- The last 2 sentences in the last paragraph of 5.1 could be reformulated for more clarity. The contrast emphasized by "while" may not be very direct when the sentences are separated.

Presentation:
- For Figure 2, though the authors listed all the languages, it's almost impossible to recognize which language specifically is represented by the dots; only the group-level information is highlighted. Thus the legend itself is barely useful, and if the authors expose some per-language analysis, it could be interesting to have a separate plot emphasizing the distribution for, for instance, one of the axis, and show whether some low-resource languages tend to be clustered with particular group of mid- or high- resource languages.

---

> ### Author Rebuttal · Authors · 2023-08-29
>
> We sincerely thank you for your insightful comments and valuable suggestions.
>
> **Answer for the Reject Reason**: Undoubtedly, the scarcity of publicly accessible data poses the primary challenge in our efforts to delve deeper into the robustness of the multilingual speech translation model. This constraint also extends to the exploration of both multi-domain capabilities and speaker identity aspects. However, we are committed to persistently working on overcoming this challenge and advancing our understanding in these areas.
>
> As for the language similarity part (Section 5), in the revised version, we aim to offer a more in-depth analysis of language similarity, with a focus on mid-resource languages within their respective language families. This decision is based on the understanding that low-resource languages do not possess a subspace of sufficiently high quality. This extended examination will be accompanied by an extra subplot depicted in Figure 2 to improve the depth of our observations, as suggested in the Presentation section.
>
> The insights drawn from Figure 2 allow us to discern the reasons behind Persian's dissimilarity to other Indo-European languages, as evidenced by its lower SVCCA scores. This observation is underpinned by the distinct location of Persian's language-specific subspace within Figure 2. Unlike other Indo-European languages, Persian's subspace aligns with a different axis, setting it apart. Meanwhile, mid-resource Indo-European languages share a common subspace with higher-resource Indo-European languages. We would add a subplot of Figure 2 focusing on Indo-European languages.
>
> **Answer for the Question**: At present, we have not conducted ablation studies based on language families or branches. Such an investigation may provide valuable insights into the behavior of multilingual speech translation. We intend to explore this aspect in future research and will present the findings in the revised version of our work. Thank you for this excellent suggestion!
>
> The typos and other issues will be corrected in the revised version. Many thanks.

---

### Official Review · Reviewer_bZDJ · 2023-08-05

**Soundness:** 3

**Excitement:**

3: Ambivalent: It has merits (e.g., it reports state-of-the-art results, the idea is nice), but there are key weaknesses (e.g., it describes incremental work), and it can significantly benefit from another round of revision. However, I won't object to accepting it if my co-reviewers champion it.

**Paper Topic And Main Contributions:**

This paper conducts an in-depth analysis of concealed representations within multilingual end-to-end speech-to-text translation (S2T) models. Notably, the authors reveal a significant departure from the knowledge transfer phenomenon observed in multilingual neural machine translation (NMT) literature, which does not extend to the realm of multilingual S2T. This conclusion is validated through a two-fold methodology. First, the authors compared translation qualities among bilingual, En->X, X->En, and X->X models. Second, they calculated similarities between sentence-level representations learned in the models via Singular Value Canonical Correlation Analysis (SVCCA). The authors showcased two interesting findings. The authors present two key insights. Firstly, the augmentation of training data proves pivotal in elevating translation quality, particularly in low-resource language pairs. With a lack of enough training data, the encoder cannot find a language-specific sub-space on top of the encoder. Secondly, the sufficiency of training data emerges as a crucial factor for capitalizing on the advantages derived from linguistic similarities across languages.

**Questions For The Authors:**

Given the results showcased in the paper, it becomes evident that when ample training data is available for all language pairs, the model acquires the ability to discern language-specific sub-spaces. This raises a pertinent question: how can we effectively strike a balance between accommodating language-specific representations and managing the overall model capacity? This becomes particularly crucial as we consider expanding the number of language directions in our endeavors.

**Reasons To Accept:**

The authors showed very interesting and useful findings through reliable analyses.

**Reasons To Reject:**

・The experiments are limited to single-domain experiments on CommonVoice, specifically within the reading speech domain. On the other hand, multilingual NMT has been explored utilizing multi-domain datasets (correct me if I'm wrong). This raises concerns regarding the broad applicability of the conclusions drawn, especially considering the potential variation when training multilingual S2T models on diverse multi-domain and multi-lingual datasets, akin to the approach employed by Whisper [Radford et al., 2022].

・Despite the authors offering two valuable insights, the paper culminates in a somewhat straightforward conclusion that data is of paramount significance. The potential for augmenting the contribution lies in the presentation of a methodology aimed at addressing the weaknesses uncovered through the conducted analyses, which could increase the overall scores.

**Reproducibility:**

4: Could mostly reproduce the results, but there may be some variation because of sample variance or minor variations in their interpretation of the protocol or method.

**Reviewer Confidence:**

5: Positive that my evaluation is correct. I read the paper very carefully and I am very familiar with related work.

---

> ### Author Rebuttal · Authors · 2023-08-29
>
> We sincerely thank you for your insightful comments and valuable suggestions.
>
> **Answer for Reject Reason 1**: Expanding the extent of these analyses to include a multi-domain context would certainly enhance their robustness, which is a consideration we intend to address in our future endeavors. However, the current feasibility of this objective is predominantly constrained by the available dataset. The MuST-C dataset, extracted from TED talks, offers a substantial corpus for speech translation, yet its limitation lies in its restricted language diversity, comprising just 8 languages. Similarly, the IWSLT SLT dataset encompasses only 3 languages. This scarcity in linguistic variety poses a challenge in comprehensively examining the results of multilingual speech translation endeavors.
>
> Effectively investigating the behavior of multilingual speech translation necessitates a broader array of languages for analysis. In this context, the CoVoST 2 dataset emerges as the most suitable candidate. Sourced from the Common Voice project, this dataset receives contributions from volunteers worldwide, thus offering a more diverse and representative range of languages for conducting multilingual speech translation tasks.
>
> Whisper's conclusion posits that "multi-domain training increases robustness and generalization." Nevertheless, the issue remains that we are confronted with a scarcity of publicly accessible multi-domain datasets. Whisper sources its data from the web, but the origins of the data remain undisclosed. Additionally, their focus is primarily on treating speech translation as a supplementary task.
>
> In summary, the notion of expanding training data to encompass multiple domains presents a promising strategy for fostering the advancement of multilingual speech translation models. However, a critical aspect for future endeavors lies in the collection of high-quality speech translation data covering various domains.
>
> **Answer for Reject Reason 2**: We are convinced that a greater availability of data, sourced from a variety of domains or tailored toward speaker identification, would enable us to explore the behavior of multilingual speech translation models in greater depth. This expanded dataset could provide valuable insights that have the potential to enhance the overall performance of multilingual speech translation systems.
>
> **Answer for the Question**: Achieving an ideal balance while including language-specific representations remains a challenging task. Some investigations in this domain show that the treatment of multilingual languages in Pre-trained Language Models (*Wu, Shijie, and Mark Dredze. "Are All Languages Created Equal in Multilingual BERT?." ACL 2020 (2020): 120.*) or Large Language Models (*Zhu, Wenhao, et al. "Multilingual machine translation with large language models: Empirical results and analysis." arXiv preprint arXiv:2304.04675 (2023).*; *Huang, Haoyang, et al. "Not All Languages Are Created Equal in LLMs: Improving Multilingual Capability by Cross-Lingual-Thought Prompting." arXiv preprint arXiv:2305.07004 (2023).*) is far from being uniform. Evidently, the issue of multilingualism in the realm of NLP remains unresolved.
>
> At present, our considerations are primarily rooted in a data-driven perspective:
>
> (a) One approach involves leveraging external ASR or MT data to pretrain the model, thereby enhancing initial states; (b) Another strategy involves data augmentation techniques such as back-translation, which recent research has shown enhances performance in bilingual speech translation.

---

### Official Review · Reviewer_RGSM · 2023-08-07

**Soundness:** 2

**Excitement:**

3: Ambivalent: It has merits (e.g., it reports state-of-the-art results, the idea is nice), but there are key weaknesses (e.g., it describes incremental work), and it can significantly benefit from another round of revision. However, I won't object to accepting it if my co-reviewers champion it.

**Paper Topic And Main Contributions:**

The paper focuses on the analysis of a multilingual speech-to-text translation model. More specifically, the paper trained a multilingual speech translation model on CoVoST 2 dataset with enhanced speech-text alignment and carried out the analysis on efficacy of multilingual training, similarity of the multilingual encoder & decoder.

**Reasons To Accept:**

1. The topic of the paper is important, given the increasing interest in multilingual speech translation in the research community. While many architectures have been proposed on the topic, there is still a need for deeper understanding of the model's behaviors. This paper broadens the discussion of the multilingual model not just on the performance (BLEU score), but also the behaviors of the model.

2. The methods used in the paper are somewhat novel. For instance, the SVCCA method is an efficient way to analyze the similarity between languages. Furthermore, the methods are model agnostic, which means it can also analyze other architectures.

3. The conclusions from the models are helpful for future study. For instance, the paper mentioned the efficacy of transferring from high-resource languages to low resource languages; improvement of audio encoding can improve the overall translation quality.

**Reasons To Reject:**

1. While the conclusion from the paper is general and seems correct from the numbers provided in the paper, the quality of the multilingual model, especially on the low resource languages, is good enough to support some of the conclusions. With half of the languages have less then 5 BLEU scores, it's not noticeably clear whether the model in fact learned meaningful representations.

In fact, this can be seen from the analysis in the similarity in section 5. In the analysis of encoder and lower-layer decoder states, the similarity in the low resource language family (e.g. Turic, Mongolic) are not as expected. They (e.g. zh, mn) have more similarity than high resource languages from another language family, indicating a potential overfitting on the high resource language (if the languages in the same language group should have higher similarity.)

2. The design of the similarity analysis can be improved. A "baseline" of the similarity analysis helps understand how multilingual capture the similarity. For instance the same SVCCA analysis on multilingual language model or speech encoder which requires looser data restriction (parallel speech/text),  One simple baseline can be run on the pure text language model and compute a similarity between two languages, and a well-trained S2T model should also capture that. The authors categorize the languages into several language families, with the indication of languages should be similar within the family. However, the conclusion is different from such indication, and we are not sure if it's because of low-quality of the model (see 1.), or it's the nature of the language (two language just not similar even in the same language group.)

2.  Missing details. For example
a. Multilingual ASR results. This will give us a better idea whether the low performance of encoder is due to the encoder.
b. The author claimed a well-aligned audio-and text helps the quality. However, the author didn't provide a multilingual setting with "bad" alignment. The conclusion is from Bilingual setting, from which the improvement could be from many other factors.
3. For Chinese and Japanese, the SacreBLEU toolkit does have tokenizers (zh and ja-mecab), and they are more widely used than character level BLEU scores.

Update after rebuttal
1. For item 1, I do acknowledge that the discrepancy of low resource language performance. What I was suggesting is, due to the low-resource setting, the representation is not well learn, and thus the analysis from it might not be informative.
2. I will remove the 3 and 4 from the reason to reject

**Reproducibility:**

4: Could mostly reproduce the results, but there may be some variation because of sample variance or minor variations in their interpretation of the protocol or method.

**Reviewer Confidence:**

4: Quite sure. I tried to check the important points carefully. It's unlikely, though conceivable, that I missed something that should affect my ratings.

---

> ### Author Rebuttal · Authors · 2023-08-29
>
> We sincerely thank you for your insightful comments and valuable suggestions.
>
> **Answer for the Reject Reason 1&2**: Research findings in multilingual NMT and multilingual language models suggest that languages within the same language family should exhibit similarity. These insights have also prompted investigations into knowledge transfer across languages in multilingual contexts. However, in the case of multilingual speech translation, our observations contradict this trend, as depicted in Figure 1. This discrepancy is likely attributed to the insufficient amount of data available for these languages, preventing the establishment of distinct language-specific subspaces.
>
> Regarding the "baseline," given the existing studies' thorough analyses of language similarity within multilingual models, particularly in multilingual NMT and multilingual LM, we have refrained from replicating this specific experiment. Instead, in the revised version, we will illustrate how multilingual models effectively capture language similarities in Section 5, enhancing readers' comprehension. Basically, a well trained S2T model also capture this trend as you conjectured. This is evident in the Romance/Germanic language branch, as illustrated in Figure 1. Nevertheless, the behavior of low-resource languages goes against this trend. To explore this further, we utilize Latent Dirichlet Allocation (LDA) for analysis. The results are shown in Figure 2, which showcases the distribution of language-specific subspaces. This distribution is influenced by the scarcity of data in low-resource languages, affecting their ability to learn these language-specific subspaces.
>
> **Answer for the Reject Reason 3**: Since the SOTA multilingual ASR models adopt seq2seq architecture, it is difficult to disentangle the encoder from the seq2seq architecture for further analyses.
>
> Table 1: Results of the multilingual ASR model which is trained on CoVoST 2 dataset.
>
> | LANGs | WER   | LANGs | WER   |
> | ----- | ----- | ----- | ----- |
> | en    | 23.14 | tr    | 47.73 |
> | fr    | 16.70 | et    | 58.83 |
> | de    | 18.94 | mn    | 60.78 |
> | ca    | 11.33 | ar    | 58.36 |
> | es    | 13.13 | cy    | 61.66 |
> | fa    | 72.88 | lv    | 48.56 |
> | it    | 20.47 | sl    | 48.28 |
> | ru    | 30.55 | sv    | 60.82 |
> | pt    | 32.63 | ta    | 73.31 |
> | zh    | 38.08 | id    | 47.39 |
> | nl    | 50.22 | ja    | 47.59 |
>
> However, we follow your suggestion to conduct the multilingual ASR experiments using transformer architecture. From these multilingual ASR results shown in Table 1, we can also observe that low-resource languages exhibit considerably lower performance levels in comparison to the high-resource languages. Additionally, when evaluating the SVCCA scores, we observe the identical pattern to that of the multilingual ST model.
>
> Regarding alignment, it becomes challenging to precisely determine the synchronization level between the audio and text modalities, given their processing through distinct modules, namely the encoder and decoder. Our conclusions predominantly arise from the comparison to bilingual models. This approach is driven by the recognition that the multilingual configuration mainly augments the text dataset while maintaining consistency in audio data.
>
> **Answer for the Reject Reason 4**: Thanks for your advice, we will use the tokenizers provided by SacreBLEU to replace char-BLEU in the revised version as shown in Table 2.
>
> Table 2: Results of Chinese and Japanese using the tokenizers provides by SacreBLEU.
>
>
> | LANGs | Bi ST (En$\rightarrow$X) | En$\rightarrow$X | X$\rightarrow$X (En$\rightarrow$X) |
> | ----- | ------------------------ | ---------------- | ---------------------------------- |
> | zh    | 23.7                     | 29.6             | 28.5                               |
> | ja    | 20.7                     | 25.2             | 24.2                               |
>
> The signatures of SacreBLEU are as follows:
>
> BLEU+case.mixed+numrefs.1+smooth.exp+tok.zh+version.1.5.1
>
> BLEU+case.mixed+numrefs.1+smooth.exp+tok.ja-mecab-0.996-IPA+version.1.5.1

---

### Meta-Review · Area_Chair_CUFS · 2023-09-18

**Recommendation:** 3

**Metareview:**

This paper presents an analysis of multilingual speech-to-text translation representations, with a focus on analyzing similarity of sentence-level multilingual representations via Singular Value Canonical Correlation Analysis (SVCCA). There is some divergence in numerical scores, but the overall tone of the review text is similar across all 3 reviews. Reviewers are enthusiastic about this type of multilingual representation analysis and its general usefulness, but they also have some concerns and suggestions for these specific experiments. In particular, two reviewers mentioned concerns about the validity of drawing conclusions regarding the similarity of languages with very small amounts of data, indicating the existing discussion on data quantity may need to be revised. The authors seem amenable to the suggested revisions, and have already provided some new results.

---

### Decision · Program_Chairs · 2023-10-07

**Decision:**

Accept-Findings

**Comment:**

This paper presents an analysis of multilingual speech-to-text translation representations, with a focus on analyzing similarity of sentence-level multilingual representations via Singular Value Canonical Correlation Analysis (SVCCA). There is some divergence in numerical scores, but the overall tone of the review text is similar across all 3 reviews. Reviewers are enthusiastic about this type of multilingual representation analysis and its general usefulness, but they also have some concerns and suggestions for these specific experiments. In particular, two reviewers mentioned concerns about the validity of drawing conclusions regarding the similarity of languages with very small amounts of data, indicating the existing discussion on data quantity may need to be revised. The authors seem amenable to the suggested revisions, and have already provided some new results.